# Telacebec (Q203)-containing intermittent oral regimens sterilized mice infected with *Mycobacterium ulcerans* after only 16 doses

**Aurélie Chauffour**[1]*, **Jérôme Robert**[1,2], **Nicolas Veziris**[1,2,3], **Alexandra Aubry**[1,2], **Kevin Pethe**[4,5], **Vincent Jarlier**[1,2]

**1** Sorbonne Université, INSERM, U1135, Centre d'Immunologie et des Maladies Infectieuses (CIMI-Paris), Paris, France, **2** Centre National de Référence des Mycobactéries et de la Résistance des Mycobactéries aux Antituberculeux, Laboratoire de Bactériologie-Hygiène, Groupe hospitalier APHP, Sorbonne Université, Site Pitié-Salpêtrière, Paris, France, **3** Département de Bactériologie, Groupe hospitalier APHP, Sorbonne Université, Site Saint-Antoine, Paris, France, **4** Lee Kong Chian School of Medicine, Nanyang Technological University, Experimental Medicine Building, Singapore, Singapore, **5** School of Biological Sciences, Nanyang Technological University, Singapore, Singapore

* aurelie.chauffour@sorbonne-universite.fr

**Data Availability Statement:** All relevant data are within the manuscript and its Supporting Information files.

## Abstract

Buruli ulcer (BU), caused by *Mycobacterium ulcerans*, is currently treated with a daily combination of rifampin and either injectable streptomycin or oral clarithromycin. An intermittent oral regimen would facilitate treatment supervision. We first evaluated the bactericidal activity of newer antimicrobials against *M. ulcerans* using a BU animal model. The imidazopyridine amine telacebec (Q203) exhibited high bactericidal activity whereas tedizolid (an oxazolidinone closely related to linezolid), selamectin and ivermectin (two avermectine compounds) and the benzothiazinone PBTZ169 were not active. Consequently, telacebec was evaluated for its bactericidal and sterilizing activities in combined intermittent regimens. Telacebec given twice a week in combination with a long-half-life compound, either rifapentine or bedaquiline, sterilized mouse footpads in 8 weeks, i.e. after a total of only 16 doses, and prevented relapse during a period of 20 weeks after the end of treatment. These results are very promising for future intermittent oral regimens which would greatly simplify BU treatment in the field.

## Author summary

The current treatment for Buruli ulcer (BU), an infection caused by *Mycobacterium ulcerans*, is based on a daily antibiotic combination of rifampin associated with streptomycin or clarithromycin. A shorter or intermittent treatment without an injectable drug would clearly simplify the management in the field. We evaluated the bactericidal activity of several new antimicrobial drugs in a mouse model of BU and found that telacebec (Q203) exhibited the greatest bactericidal effect. We subsequently identified new antibiotic combinations containing telacebec with high sterilizing activity when administered twice a week for 8 weeks, i.e. at a total of only 16 doses.

**Funding:** This work was funded by the Raoul Follereau Foundation. The funders had no role in study design, data collection and analysis, decision to publish, or preparation of the manuscript.

**Competing interests:** The authors have declared that no competing interests exist.

## Introduction

Buruli ulcer (BU), caused by *Mycobacterium ulcerans*, was only treated by surgery until 2004 [1]. The first medical treatment recommended by the World Health Organization (WHO) was an eight-week daily treatment based on an association of two antibiotics, rifampin (RIF), an oral ansamycin, and streptomycin (STR), an injectable aminoglycoside [1]. Currently a promising fully oral regimen combining RIF and clarithromycin (CLR), a macrolide compound [2,3], is tested clinically on a large scale (NCT01659437, clinicaltrials.gov).

The oral RIF-CLR combination was promoted to eliminate the toxic effects and injections of aminoglycosides, resulting in greater patient adherence and safety. Nevertheless, this combination is given daily for eight weeks. Shorter or intermittent treatment would facilitate adherence as well as supervision by healthcare workers. For instance, many Buruli ulcer patients with small-to-moderate size wounds are on ambulatory care and visit healthcare centres twice or three times per week for dressing changes, a rhythm that could allow receiving supervised intermittent antibiotic administration [4].

Our main objective was to identify alternative oral regimens active against BU by using a validated BU animal model. As a first step, we screened several new drug candidates for their *in vivo* bactericidal activity against *M. ulcerans*. Based on available data on their activity against *M. tuberculosis* or *M. ulcerans*, the following compounds, with either short or long half-life, were selected as potentially interesting: selamectin (SEL) and ivermectin (IVE), two drugs from the avermectin family with antiparasitic properties [5–7]; tedizolid (TDZ) [8], a new oxazolidinone active against *M. tuberculosis* that has the same mechanism of action as linezolid (LZD), a drug active against *M. tuberculosis* and *M. ulcerans* [9] with high bioavailability and a long half-life [10]; the 2-piperazino-benzothiazinone 169 (PBTZ), shown to be highly active against *M. tuberculosis* [11,12]; the imidazopyridine amine telacebec (Q203) that targets the respiratory terminal oxidase cytochrome bc1:aa3 in *M. tuberculosis* and that was recently shown to prevent mortality and reduce CFU counts in the footpads of mice infected with *M. ulcerans* [13] and to exhibit sterilizing activity when administrated in combination with other drugs [14].

Because the first-step screening demonstrated that Q203 was the compound with the highest bactericidal activity, and because of the long half-life of this compound [15], we measured, in a second experiment, the bactericidal and sterilizing activity of intermittent regimens containing Q203 combined with rifapentine or bedaquiline, other antibiotics with a long half-life and known to be active against *M. ulcerans* [16,17].

## Methods

### Infection of mice with *M. ulcerans*

In the first and the second experiment, respectively, 190 and 390 4-week-old female BALB/c/j mice were used (Janvier Labs, Le Genest Saint-Isle, France). Mice were inoculated, according to Shepard [18] in the left hind footpad with 0.03 ml of a bacterial suspension containing 5 $\log_{10}$ bacilli of the *M. ulcerans* strain Cu001. The number of live bacilli in the bacterial suspension was determined to be 5.02 and 4.6 $\log_{10}$ in the first and second experiment, respectively, by culturing the inoculum on Lowenstein-Jensen (LJ) medium. This strain, isolated in 1996 from a BU patient in Adzopé, Ivory Coast [19], was kindly provided by the local laboratory, blinded to patient identity. The strain is susceptible to all drugs used in BU treatment and was maintained in our laboratory through regular mouse footpad passage.

### Treatment of mice

Treatment was initiated when the infection was well established, *i.e.* at a footpad swelling index of "2" (inflammatory swelling limited to the inoculated footpad) to "3" (inflammatory swelling involving the entire inoculated footpad) on a 4-grade scale [19] (Fig 1). This stage of infection was reached six weeks after the inoculation. Mice were randomly allocated into eight groups (1st experiment) and ten groups (2nd experiment) using a randomization table generated by the web site Randomization.com (https://www.randomization.com).

The groups were treated (with respect to drug, dosage, as well as number of doses/number of weeks, respectively) as follows:

In the first experiment, we used 8 groups including one untreated control group of 30 mice and seven antibiotic-treated groups of 20 mice, each treated with either TDZ, 10 mg/kg, 5/7; LZD, 100 mg/kg, 5/7; SEL, 12 mg/kg, 1/7; IVE, 1 mg/kg, 5/7; Q203, 5 mg/kg, 5/7 or PBTZ, 25 mg/kg, 5/7 and, as treatment control, RIF, 10 mg/kg, 5/7 alone or combined with STR, 150 mg/kg.

In the second experiment, out of 10 groups in total, we used 6 groups including one untreated control group of 27 mice and five antibiotic-treated groups of 27 mice, each receiving monotherapy with RIF, 10 mg/kg, 5/7; rifapentine (RPT), 20 mg/kg, 2/7; bedaquiline (BDQ), 25 mg/kg, 2/7; or Q203, 5 mg/kg, 5/7 or 2/7 while we used 4 groups of 57 mice each treated with drug combinations (dosages as above), i.e. Q203-RIF, 5/7; Q203-RPT, 2/7; Q203-BDQ, 2/7 or RIF-CLR, 100 mg/kg, 5/7 as control.

LZD was purchased from Pfizer, France; SEL and IVE from Merck, France; RIF from Sandoz, France; STR from Panpharma, France; and CLR from Abbott, France. TDZ was kindly

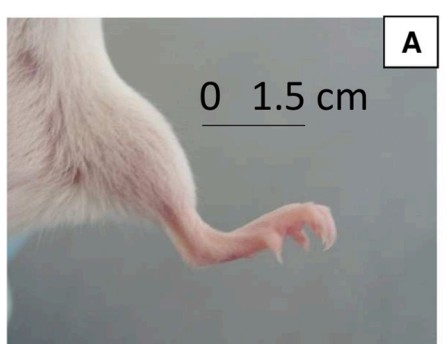
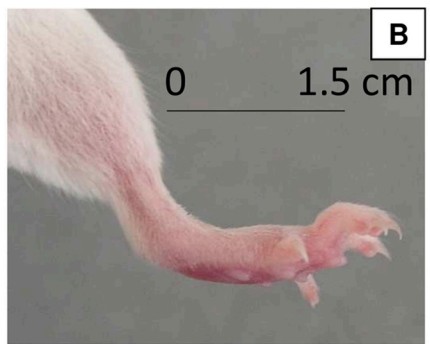
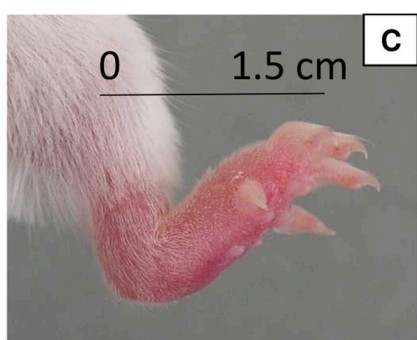
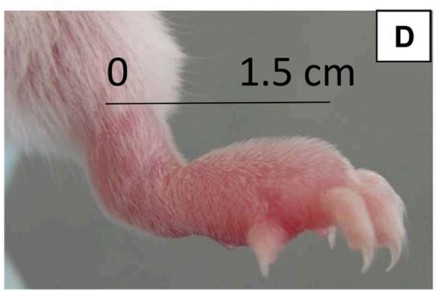
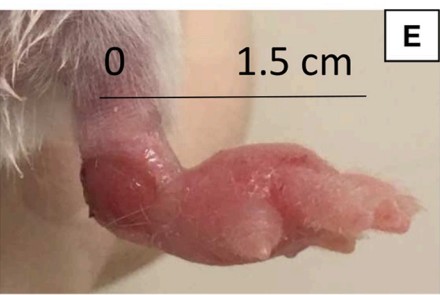

**Fig 1.  Images representative of the lesion indexes used** (A) index 0; (B) index 1; (C) index 2; (D) index 3; (E) index 4. Measure of the index was taken at the middle of the footpad (see the black bracket). Measures of indexes were as follows: 2mm for index 0; 3mm for index 1; 4.5mm for index 2; 5.5mm for index 3; and 6.5mm for index 4.

provided by MSD-MERCK group, BDQ by Janssen Pharmaceutica, Belgium, and PBTZ by Stewart Cole (Global Health Institute, Ecole Polytechnique Fédérale de Lausanne, Lausanne, Switzerland; Institut Pasteur, rue du Docteur Roux, France). Q203 was custom-synthetized at GVKBio (GVK Biosciences Private Limited, Hyderabad, India) and contributed by Kevin Pethe.

Antibiotics were suspended in 0.05% agar-containing distilled water, except for STR, which was dissolved in buffered saline, Q203 in 1% DMSO-20% TPGS (D-α-Tocopherol polyethylene glycol 1000 succinate) and PBTZ in a solution of 1% carboxyl-methylcellulose-1% Tween 80. BDQ was provided by Janssen Pharmaceutica in a 20% hydropropyl-β-cyclodextrin formulation.

Treatments were administered for 4 or 8 weeks in both experiments and all drugs were orally administered by gavage in a final volume of 0.2 ml, except STR, which, in the same volume, was injected subcutaneously.

### Assessment of *M. ulcerans* infection and effectiveness of treatment

Two methods were used for assessing the development of *M. ulcerans* infection and the effectiveness of treatments: (i) a clinical method consisting in weekly evaluation of the lesion index as previously described [19] and (ii) a bacteriological method consisting in the cultivation of *M. ulcerans* from the mouse footpads. When footpad swelling totally regresses under treatment, some redness often remains, indicative of scar tissue. We chose to assign index "1" to this stage, as index "0" is assigned to a normal footpad. For CFU enumeration in footpads, mice were sacrificed by cervical dislocation as recommended by the European directive 2010/63 and the French decree n˚2013–118. Mouse footpads were removed aseptically and ground in 2 ml of Hank's balanced salt solution in a tissue grinder (Octo Dissociator GentleMACS, Miltenyi Biotec, France). Suspensions were then plated onto LJ slants containing vancomycin (10 μg/ml), colistin (40 μg/ml) and amphotericin B (10 μg/ml) to limit contamination. In the first experiment, for the untreated group and the groups treated with RIF, TDZ, LZD, SEL, IVE or PBTZ, suspensions were serially 10-fold diluted to $10^{-4}$, and 0.1 ml of the dilutions was plated in duplicate onto LJ media. For the Q203 and the RIF-STR-treated groups, the entire volume of the footpad suspensions was plated onto LJ-media, at 0.2 ml per slant. In the second experiment, suspensions were serially diluted 10-fold to $10^{-4}$ for the untreated group, to $10^{-3}$ for the RIF and RPT groups at 2, 4 and 8 weeks and for the BDQ group at 2 weeks; to $10^{-2}$ for the Q203 2/7, Q203 5/7 and RIF-CLR groups at 2 weeks, and for the BDQ group at 4 weeks. A volume of 0.1 ml of each of these serial dilutions was plated in duplicate onto LJ media. For all the remaining time points, including those for all Q203 combinations, the entire volume of the footpad suspensions was plated onto LJ media, at 0.2 ml per slant. All tubes were incubated at 30˚C for 90 days.

In the first experiment, the footpad lesion indexes were determined weekly during the 8-week period of treatment and CFUs were counted at weeks 4 and 8. In the 2nd experiment, footpad lesion indexes were determined weekly during the 8-week period of treatment and CFUs were counted at weeks 2, 4 and 8. Moreover, in the latter experiment, 30 mice that had been treated for 8 weeks with combination therapies were held without treatment during an additional period of 20 weeks to monitor relapses of *M. ulcerans* infection. Lesion indexes were determined weekly during this period and CFUs were counted at the end of the observation period.

### MIC determination

In order to assess a possible acquisition of resistance to the antibiotics used during treatment, MICs of the antibiotics used for the treatment were determined for the bacilli isolated from

relapsing mice during the observation period and for the initial strain Cu001. The isolates were suspended in distilled water to match the turbidity of a Mc Farland 3 standard. RIF and CLR were tested on Middlebrook 7H11 + 10% OADC (Oleicacid-Albumin-Dextrose-Catalase) medium (pH 7.4) and CLR was also tested on Mueller Hinton medium (pH 6.6). RIF was dissolved in dimethylformamide and CLR in distilled water, and both were twofold diluted in their respective solvent and incorporated into the culture media to obtain final concentrations ranging from 4 to 0.12 μg/ml. Suspensions (0.1 ml) of two distinct bacillus isolates (pure and diluted $10^{-2}$) were plated onto drug-containing and drug-free media used for growth control. All media were incubated at 30°C and examined after 60 and 90 days. The minimum inhibitory concentration (MIC) was defined as the lowest concentration of the drug preventing ≥99% of the growth on drug-free medium [20].

## Statistical analysis

The Mann–Whitney *U* test was used to analyze the results (BiostaTGV). Relapse rates were also compared using Fisher's exact test. A p-value <0.05 was considered to be statistically significant. A significant decrease in the mean CFU value per footpad of treated as compared to untreated mice was used to assess the bactericidal effect of a given drug regimen.

## Ethics statement

The experimental project was favorably evaluated by the ethics committee n°005 Charles Darwin localized at the Pitié-Salpêtrière Hospital and clearance was given by the French Ministry of Education and Research under the number APAFIS#9576–20170301171176185 v2. Our animal facility received the authorization to carry out animal experiments (license number C-75-13-01) on April 27, 2017. The persons who carried out the animal experiments had followed a specific training recognized by the French Ministry of Education and Research.

# Results

## First experiment

The mean lesion index (MLI) was 2.8±0.66 at the start of treatment (**Fig 2**). Untreated control mice with swollen footpads and MLIs that increased from 2.8 to 4 after 2 weeks, and mice with extensive lesions (inflammatory swelling of the whole limb, accompanied by a deteriorating general condition) either died or had to be sacrificed at week 4. MLIs in RIF-treated control groups increased to 3.8 after one week and then decreased to remain stable at 3.4, while decreasing to 2 in the RIF-STR group. MLIs in the SEL-, IVE-, TDZ- and PBTZ-treated groups continued to increase and mice either died or had to be sacrificed at week 4 due to advanced lesions comparable to those observed in untreated mice. In contrast, the MLI in the Q203-treated group decreased to 1.9 after 1 week of treatment and to 1.2 after 8 weeks.

All untreated mice had culture-positive footpads at the beginning of treatment, with a mean of 6.93 ± 0.20 $\log_{10}$ CFUs, a count that remained stable at week 4 (**Table 1**). All mice remained culture-positive after 4 weeks of treatment in the RIF control groups but the mean CFU counts were significantly lower ($p$ <0.05) than those in the untreated mouse group (4.58 ± 1.29 $\log_{10}$ for the RIF-($p$ = 0.0007) and 2.15 ± 1.20 $\log_{10}$ for the RIF-STR ($p$ = 0.000006) treated groups, respectively). After 8 weeks of treatment, only 5 out of 8 mice in the RIF groups remained culture-positive with a low mean CFU count (<1 $\log_{10}$). After 4 weeks of treatment, all mice were culture-positive in the TDZ, SEL, IVE and PBTZ groups, with CFU counts not statistically different from those in the untreated control group. Due to advanced footpad lesion and mortality of mice, the evaluation point initially planned at week 8 was cancelled for

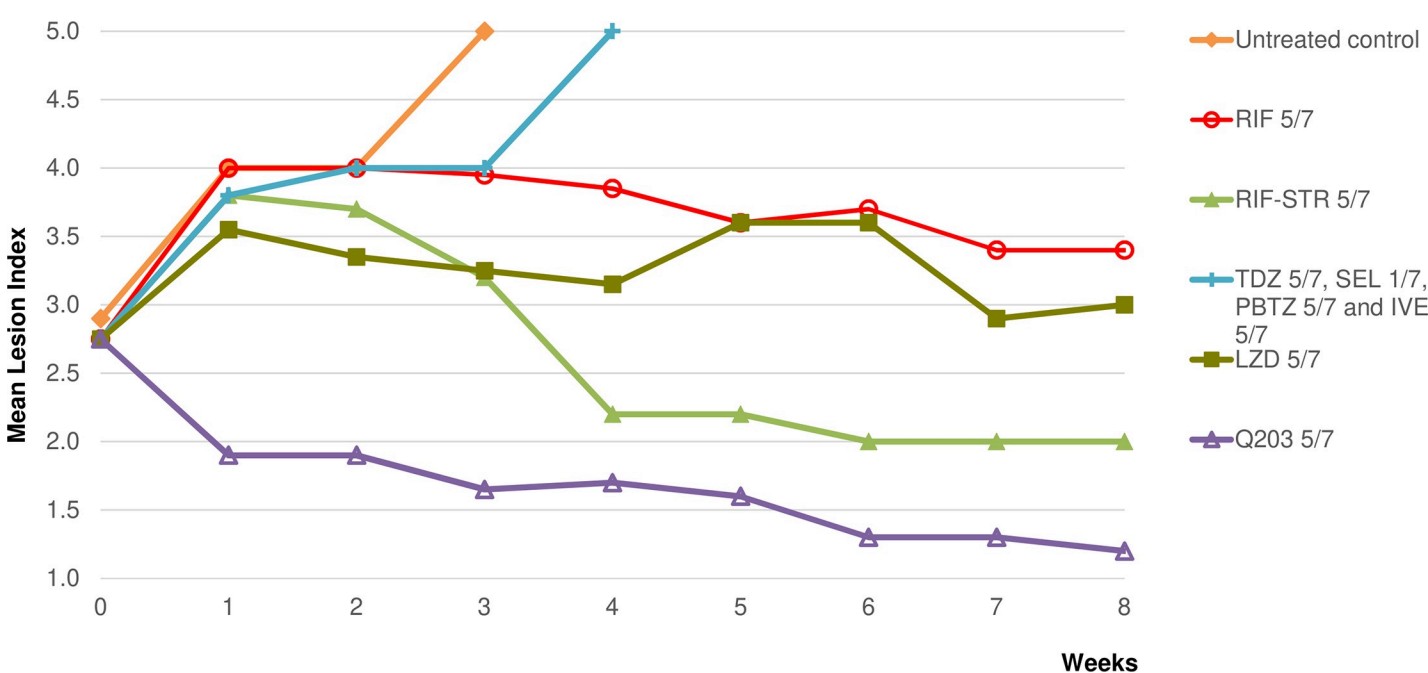

**Fig 2. First experiment: Mean lesion index of *M. ulcerans*-infected mouse footpads during treatment for 8 weeks.** Doses were as follow: rifampin (RIF), 10 mg/kg; streptomycin (STR), 150 mg/kg; tedizolid (TDZ), 10 mg/kg; linezolid (LZD), 100 mg/kg; selamectin (SEL), 12 mg/kg; ivermectin (IVE), 1 mg/kg; telacebec (Q203), 5 mg/kg; PBTZ169 (PBTZ), 25 mg/kg. The number of doses/number of weeks for RIF alone were 5/7; for RIF-STR, 5/7; for TDZ, 5/7; for LZD, 5/7; for SEL, 1/7; for IVE, 5/7; for Q203, 5/7 and for PBTZ, 5/7.

these groups. After 4 weeks of treatment, 3 out of 10 mice were culture-negative in the Q203-treated group with a low mean CFU count ($1.14 \pm 1.30 \log_{10}$), i.e. different from those in the RIF-treated group ($p = 0.00002$). After 8 weeks of treatment, all mice were culture-negative in the Q203 group.

## Second experiment

The MLI was $3\pm0.79$ at the start of the treatment (**Fig 3**). The footpad MLIs of untreated control mice increased from 3 to 4 after 2 weeks and the mice had to be sacrificed at week 4 due to extensive BU lesions. The MLIs in the RIF-, RPT- and BDQ-treated groups increased to 3.8–4 after one week of treatment and then decreased, by week 8, to 3.6 (BDQ), 3 (RIF) and 2.7 (RPT). In the Q203-treated groups (*n* of doses/*n* of weeks, 5/7 or 2/7), MLIs increased slightly to 3.5 after one week of treatment and then decreased rapidly to 1.6–1.7 at week 8. The MLI in the group treated with RIF-CLR increased to 4 after 1 week of treatment, then decreased to 1.4 at week 12 and remained stable until week 20, but increased again to 2.6 at week 28. The MLI in the group treated with Q203-BDQ increased slightly to 3.6 after one week of treatment and decreased thereafter to 1.4 at week 12, a level that remained stable until week 28. The MLIs in the groups treated with Q203 combined with RIF or RPT decreased rapidly to 1.2–1.3 at week 8 and remained at that level until week 28.

All untreated mice had culture-positive footpads at the beginning of treatment, with a mean of $6.87 \pm 0.10 \log_{10}$ CFUs, a count that remained stable at weeks 2 and 4 (**Table 2**). All treated mice remained culture-positive after 2 weeks of treatment but the CFU counts were significantly lower in all treated groups compared to those in the untreated group. Moreover, CFU counts were lower in the RPT (compared to RIF $p = 0.008$ and $p = 0.0003$ compared to BDQ), Q203 (for Q203 2/7 compared to RIF $p = 0.008$; compared to BDQ $p = 0.001$; for Q203

**Table 1. First experiment: Results of footpad cultures during the treatment of mice infected with *M. ulcerans*.**

| Regimen[a] (*n* doses/*n* weeks) | Results during treatment | | | | | |
| --- | --- | --- | --- | --- | --- | --- |
| | Day 0 | | Week 4 | | Week 8 | |
| | Culture positivity rate | Mean (±SD) CFU per group | Culture positivity rate | Mean (±SD) CFU per group | Culture positivity rate | Mean (±SD) CFU per group |
| **Untreated control** | 10/10 | 6.93±0.20 | 13/13[b] | 6.81±0.36 | | |
| **RIF 5/7** | | | 9/9[c] | 4.58±1.29 | 5/8[c] | 0.47±1.16 |
| **RIF-STR 5/7** | | | 8/8[d] | 2.15±1.20 | 3/9[d] | 0.57±0.90 |
| **TDZ 5/7** | | | 7/7[e] | 6.35±0.59 | [f] | |
| **LZD 5/7** | | | 10/10 | 2.83±0.62 | 9/10 | 1.87±1.11 |
| **SEL 1/7** | | | 4/4[g] | 6.24±0.82 | [f] | |
| **IVE 5/7** | | | 8/8[h] | 6.30±0.82 | [f] | |
| **Q203 5/7** | | | 7/10 | 1.14±1.30 | 0/10 | |
| **PBTZ 5/7** | | | 7/7[i] | 7.08±0.64 | [f] | |

[a]: treatment began 6 weeks after inoculation with 5.02 log$_{10}$ bacilli per footpad when the infected swollen footpads had reached a lesion index of 2.8.

Drugs were administered 5-times a week, except for selamectin (SEL) which was administered once a week. Dosages were as follows: rifampin (RIF), 10 mg/kg; streptomycin (STR), 150 mg/kg; tedizolid (TDZ), 10 mg/kg; linezolid (LZD), 100 mg/kg; SEL, 12 mg/kg; ivermectin (IVE), 1 mg/kg; telacebec (Q203), 5 mg/kg; PBTZ169 (PBTZ), 25 mg/kg.

[b]: due to advanced lesions, all mice from the untreated control group were in fact sacrificed at week 3; nevertheless, footpad cultures were contaminated in 7 out of 20 mice.

[c]: 3 mice died in the RIF groups due to an accident during gavage.

[d]: 3 mice died in the RIF-STR groups due to an accident during gavage.

[e]: footpad cultures were contaminated due to advanced lesions in 3 mice.

[f]: cultures at week 8 were not performed because of advanced necrotized footpad lesions.

[g]: 5 mice died from *M.ulcerans* infection and footpad cultures were contaminated due to advanced lesion in 1 mouse.

[h]: 1 mouse died from *M.ulcerans* infection and footpad cultures were contaminated due to advanced lesion in 1 mouse.

[i]: footpad cultures were contaminated due to advanced lesions in 3 mice.

5/7 compared to RIF $p = 0.01$ and $p = 0.0001$ compared to BDQ) and the 4 combined-treatment groups than in the RIF and BDQ groups (p-values for combined treatment groups vs RIF: CR $p = 0.003$; QR $p = 0.002$; QB $p = 0.006$; QP $p = 0.002$; all p-values for combined treatment groups vs BDQ were 0.0002). After 4 weeks of treatment, some of the mice became culture-negative, especially in the groups treated with Q203-RPT and Q203-BDQ in which the CFU counts were very low (~0.2 log$_{10}$). CFU counts at 4 weeks remained lower in the RPT and Q203 than in the RIF ($p = 0.006$, $p = 0.04$ and $p = 0.002$ for RPT, Q203 2/7 and Q203 5/7 respectively) and BDQ groups ($p = 0.007$, $p = 0.001$ and $p = 0.001$ for RPT, Q203 2/7 and Q203 5/7 respectively) and they were lower in the Q203-RPT ($p = 0.04$ and $p = 0.03$) and Q203-BDQ ($p = 0.03$ and $p = 0.01$) groups than in the Q203-RIF and RIF-CLR groups. After 8 weeks of treatment, all mice treated with Q203 alone or combined with RIF (5/7) or RPT (2/7) or BDQ (2/7) became culture-negative. Few mice remained culture-positive in the RIF, RPT and RIF-CLR groups with very low mean CFU counts (<0.5 log$_{10}$) but all remained culture-positive in the BDQ group (1.28 ± 0.93 log$_{10}$).

During the 20-week observation period after stopping the treatment, no clinical or bacteriological relapse was observed in the 3 groups treated with Q203 combinations whereas 8 out of 26 mice treated with RIF-CLR relapsed with a mean CFU count of 0.87 ± 0.93 log$_{10}$. This proportion was significantly higher than those in other groups ($p = 0.001$).

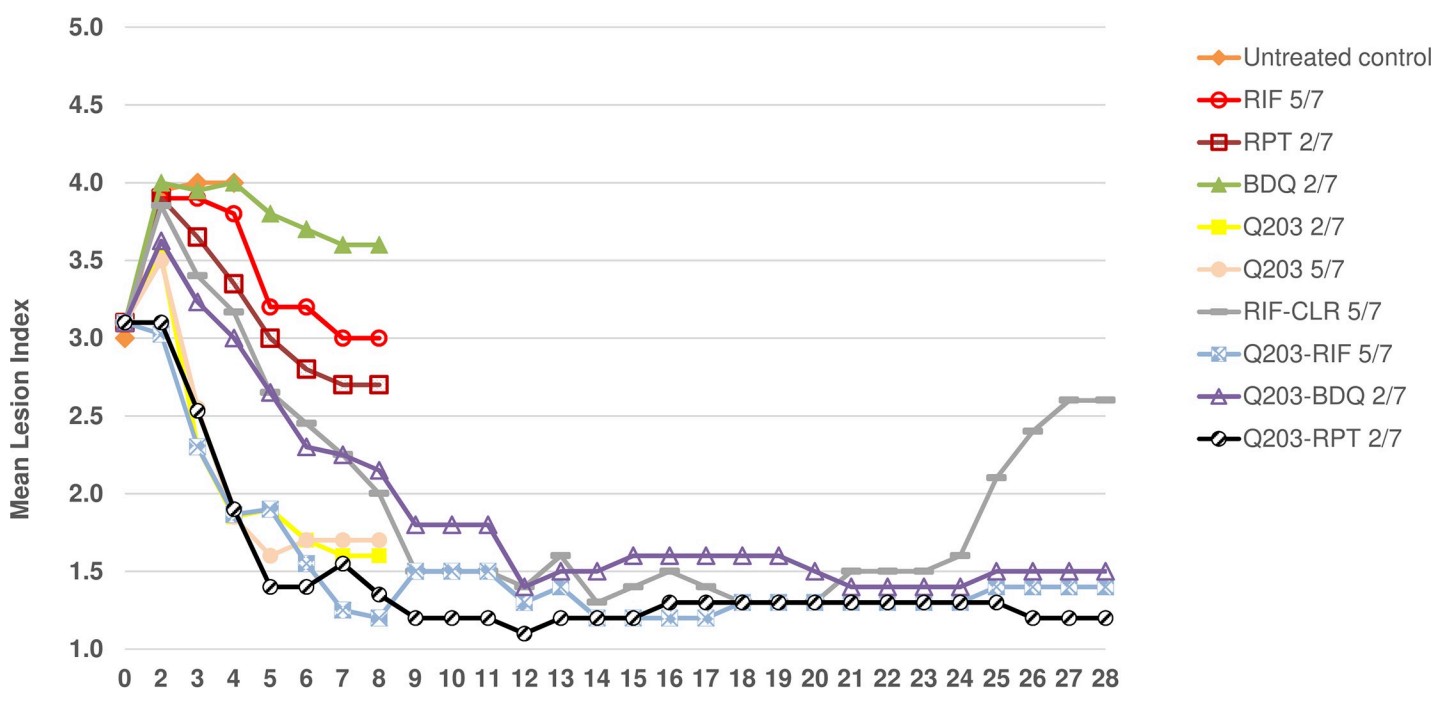

**Fig 3. Second experiment: Mean lesion index of M. ulcerans-infected mouse footpads during and after treatment for 8 weeks.** Dosages were as follows: rifampin (RIF), 10 mg/kg; rifapentine (RPT), 20 mg/kg; bedaquiline (BDQ), 25 mg/kg; Q203, 5 mg/kg; clarithromycin (CLR), 100 mg/kg. RIF was given (n of doses/n of weeks) 5/7; RPT, 2/7; BDQ, 2/7; Q203, 2/7 or 5/7; RIF-CLR, 5/7; Q203-RIF, 5/7; Q203-BDQ, 2/7 and Q203-RPT, 2/7. 1.2–1.3 at week 8 and remained at that level until week 28.

## MICs for *M. ulcerans* bacilli recovered from relapsing mice

The MICs for bacilli isolated from the 8 relapsing mice in the RIF-CLR-treated group remained stable when compared to the initial MICs of RIF (0.5–1 µg/ml) and CLR (0.5 µg/ml) found for *M. ulcerans* Cu001 (the MICs of the latter were found to be the same on the 7H11 and MH media).

## Discussion

Although nearly all Buruli ulcer lesions can be successfully treated by a two-month antibiotic combination regimen administered daily [21,22], new shorter and/or intermittent regimens would greatly simplify the treatment procedure in the field. Indeed, intermittent treatment would allow for organizing direct supervision of therapy and for combining drug administration and change of dressings in remote places where BU is prevalent.

The key finding of the present work was that regimens combining telacebec with rifapentine or bedaquiline, two long lasting drugs, sterilized infected mouse footpads after twice weekly administration during for 8 weeks, i.e. after a total of only 16 doses.

Our first *in vivo* screening experiment aimed at identifying newer bactericidal drugs. Indeed, available data on the activity of several new drugs against *M. tuberculosis* or *M. ulcerans* justified a systematic evaluation in a BU mouse model that has been successfully used for many years for this purpose [9]. IVE, SEL and TDZ were not bactericidal after 4 weeks of treatment and failed to prevent mortality during 8 weeks. The doses used in our experiment were taken from available pharmacokinetic data. IVE, a long-lasting drug in humans (half-life, 15-19h) and in mice (9h), when used in mice at a dose of 0.2 mg/kg, was shown to yield serum

**Table 2. Second experiment: Results of footpad cultures during the treatment of mice infected with *M. ulcerans* (for 2, 4 or 8 weeks) and relapse rate after treatment completion.**

| Regimen[a] (n doses/n weeks) | Results during treatment | | | | | | | | Results after observation period | |
|---|---|---|---|---|---|---|---|---|---|---|
| | Day 0 | | Week 2 | | Week 4 | | Week 8 | | Week 28 | |
| | Culture positivity rate | Mean (±SD) CFU per group | Culture positivity rate | Mean (±SD) CFU per group | Culture positivity rate | Mean (±SD) CFU per group | Culture positivity rate | Mean (±SD) CFU per group | Culture positivity rate | Mean (±SD) CFU per group |
| **Untreated control** | 9/9 | 6.87±0.10 | 8/8[b] | 7.00±0.16 | 6/6[b] | 6.84±0.35 | | | | |
| **RIF 5/7** | | | 9/9 | 5.71±0.65 | 6/6[c] | 2.78±0.81 | 2/8[c] | 0.44±0.81 | | |
| **RPT 2/7** | | | 8/9 | 4.18±1.71 | 3/9 | 0.63±1.03 | 1/9 | 0.20±0.59 | | |
| **BDQ 2/7** | | | 7/7[d] | 6.12±0.31 | 7/7[d] | 5.48±0.14 | 5/5[d] | 1.28±0.93 | | |
| **Q203 2/7** | | | 9/9 | 4.88±0.42 | 5/9 | 1.42±1.21 | 0/9 | | | |
| **Q203 5/7** | | | 9/9 | 4.90±0.29 | 7/9 | 0.74±0.79 | 0/9 | | | |
| **RIF-CLR 5/7** | | | 9/9 | 4.42±0.55 | 6/7[e] | 1.15±0.84 | 1/8[e] | 0.22±0.63 | 8/26[e] | 0.87±1.36 |
| **Q203-RIF 5/7** | | | 9/9 | 4.52±0.36 | 9/9 | 1.20±1.04 | 0/8[f] | | 0/30 | |
| **Q203-BDQ 2/7** | | | 9/9 | 4.65±0.75 | 2/9 | 0.17±0.51 | 0/9 | | 0/30 | |
| **Q203-RPT 2/7** | | | 9/9 | 4.40±0.78 | 2/9 | 0.25±0.49 | 0/9 | | 0/30 | |

[a]: treatments were begun 6 weeks after inoculation of 4.6 $\log_{10}$ CFUs per footpad when the infected swelling footpads reached a lesion index of 3. Drugs were administered two or five times a week and dosages were as follows: rifampin (RIF), 10 mg/kg; rifapentine (RPT), 20 mg/kg; bedaquiline (BDQ), 25mg/kg; telacebec (Q203), 5 mg/kg and clarithromycin (CLR), 100 mg/kg.

[b]: footpad cultures were contaminated due to advanced lesion in 1 mouse in the untreated Week 2 control group, and in 1 mouse in the Week 4 group.

[c]: footpad cultures were contaminated due to advanced lesions in 3 mice in the Week 4 RIF group, and in 1 mouse in the Week 8 group.

[d]: footpad cultures were contaminated due to advanced lesions in 1 mouse in the Week 2 BDQ group, in 2 mice in the Week 4 group and in 2 mice in the Week 8 group; 2 mice died due to an accident during gavage in the Week 8 group.

[e]: footpad cultures were contaminated due to advanced lesion in 2 mice in the Week 4 RIF-CLR group and in 3 mice in the relapse observation group; 1 mouse each died due to an accident during gavage in the Week 8 group and in the relapse observation group.

[f]: 1 mouse died due to an accident during gavage in the Week 8 Q203-RIF group.

concentrations lower than those found in humans at standard therapeutic dosage [7]. We therefore used a higher dose (1 mg/kg), which also failed to control infection in our model. A similarly unfavorable result was obtained with SEL used at 12 mg/kg as proposed in a previous publication [23]. However, it has been suggested that these two avermectin compounds might be used safely at even higher doses [7], which could be evaluated in future studies. The dose of TDZ used in the present study, i.e. 10 mg/kg, was shown to yield pharmacokinetic profiles close to those observed in humans at the therapeutic dose of 200 mg [24–26]. Contrasting with the disappointing results obtained with TDZ, and as reported in a previous study [9], a marked bactericidal activity was obtained with LZD, an oxazolidinone included in the experiment as a positive control for TDZ. Surprisingly, PBTZ was not bactericidal in our BU model at 25 mg/kg, a dose shown to be active against *M. tuberculosis* in mice [11]. Yet in *M. ulcerans*, as well as in *M. tuberculosis*, DprE1, the target of benzothiazinones, carries a cysteine at position 387 in lieu of a serine or an alanine, which have been shown to confer natural resistance to PBTZ in *M. avium* or *M. aurum*, respectively [27]. Thus, the reason for the disappointing result obtained with PBTZ in our BU model is unclear.

RIF and RIF-STR were highly bactericidal as observed in all our preceding studies [2,28]. Q203 drastically reduced the lesion index and CFU counts after 4 weeks of treatment and all

mice became culture-negative after 8 weeks. These results obtained with Q203 were clearly superior to those obtained with the usual positive control with RIF alone, and even with the reference combination regimen RIF-STR.

Drugs with low bactericidal activity, when given as monotherapy, might be of interest when used in combination with other drugs. However, since our goal was to obtain the most effective combination regimen, we selected, for the second experiment, combinations of drugs shown to be highly active separately. The results of this experiment demonstrated that regimens combining Q203 with RIF or RPT or BDQ were not only bactericidal, making all footpads culture-negative after 8 weeks of treatment, but also sterilized them and prevented relapse during an observation period of 20 weeks after stopping the treatment. These promising results were obtained after administering twice weekly for 8 weeks, i.e. after a total of only 16 doses, the combinations of Q203 with either RPT or BDQ. These three drugs are long-lasting, with serum half-lives in mice after a single dose being Q203 23 h [15]; RPT 25 h [16] and BDQ 53 h [17]. Recently, regimens combining RPT with CLR or BDQ, administered twice weekly for 8 weeks, were found to be as bactericidal and as sterilizing as a daily RPT-CLR regimen [28].

We recently showed that Q203 was very active in the footpads of mice infected with M. *ulcerans* [13]. Lately, a study by Converse *et al* confirmed this finding and furthermore showed that triple combinations of Q203 administered at the higher dose of 10 mg/kg, either with RPT and clofazimine, RPT and BDQ or BDQ and clofazimine, as well as a quadruple combination of these four drugs, were sterilizing after 2 weeks of daily treatment in a BU animal model [14], leading to the conclusion that targeting the *M. ulcerans* respiratory chain with several drugs is an effective strategy for designing new shortened treatments of BU. Thus, Q203 appears clearly an important candidate for future treatment of BU, either in daily or intermittent regimens.

The fact that, in the present work, few bacilli were still found by culture in 1 out of 8 mice after 8 weeks of treatment with RIF-CLR 5/7, and that 8 out of 26 mice relapsed with low CFU counts within 20 weeks after the end of this regimen, was surprising since bactericidal and sterilizing activity of RIF-CLR was shown in our previous studies. Other authors have observed, 20 weeks after stopping treatment with RIF-CLR 5/7, 25% of relapses based on recurrent swelling or ectopic lesions [29]. The susceptibility to RIF and to CLR of the bacilli isolated from relapsing mice in our study was unchanged, ruling out the selection of resistant mutants during treatment. Relapses could be explained by unusually high bacterial loads reached in the present work when compared to those in our previous studies, *i.e.* 3–4 times higher at the start of treatment and 4–10 times higher after 2–4 weeks in untreated mice. Nevertheless, this fact strengthens further the good results obtained with the Q203-containing combination regimens.

New drugs are rare for the treatment of BU. The last active new marketed drug was BDQ, initially assessed with success for tuberculosis and found later to be very active against *M. ulcerans* in a BU animal model [28]. Therefore, the excellent *in vivo* activity of Q203 against *M. ulcerans* constitutes a step forward. In a recent study, the basis of this promising result was found to be reductive evolution in most strains of this species that led to hyper susceptibility to Q203 by elimination due to the absence of alternate terminal electron acceptors, thereby making the target, respiratory cytochrome *bc*1:*aa*3, crucial for survival [13].

Whereas, triple or quadruple combinations of Q203 at 10mg/kg, administered with RPT, clofazimine and BDQ were found to sterilize infected mice after 2 weeks of daily treatment, we demonstrated in the present study that double combinations of Q203 at 5 mg/kg with either RPT or BDQ, were also sterilizing after 16 doses of a twice weekly treatment, due to the long half-life and good bioavailability of these three drugs. The latter intermittent scheme of oral treatment with two drugs would greatly simplify the treatment of BU on ambulatory care in

the field, when patients living in remote areas visit healthcare centers twice or three times per week for dressing changes, a rhythm that could allow receiving supervised intermittent antibiotic administration.

## Author Contributions

**Conceptualization:** Aurélie Chauffour, Vincent Jarlier.

**Formal analysis:** Jérôme Robert, Vincent Jarlier.

**Funding acquisition:** Jérôme Robert, Nicolas Veziris, Alexandra Aubry, Vincent Jarlier.

**Methodology:** Aurélie Chauffour, Jérôme Robert, Vincent Jarlier.

**Project administration:** Alexandra Aubry.

**Resources:** Aurélie Chauffour.

**Software:** Aurélie Chauffour.

**Supervision:** Vincent Jarlier.

**Validation:** Jérôme Robert, Kevin Pethe, Vincent Jarlier.

**Writing – original draft:** Aurélie Chauffour, Vincent Jarlier.

**Writing – review & editing:** Jérôme Robert, Nicolas Veziris, Alexandra Aubry, Kevin Pethe.

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
