## [Decision Letter · Decision Letter 0]

20 Dec 2019

Dear Mrs Chauffour:

Thank you very much for submitting your manuscript "Q203 containing fully intermittent oral regimens exhibited high sterilizing activity against Mycobacterium ulcerans in mice" (#PNTD-D-19-01766) for review by PLOS Neglected Tropical Diseases. Your manuscript was fully evaluated at the editorial level and by independent peer reviewers. The reviewers appreciated the attention to an important problem, but raised some substantial concerns about the manuscript as it currently stands. These issues must be addressed before we would be willing to consider a revised version of your study. We cannot, of course, promise publication at that time.

We therefore ask you to modify the manuscript according to the review recommendations before we can consider your manuscript for acceptance. Your revisions should address the specific points made by each reviewer. 

When you are ready to resubmit, please be prepared to upload the following:

(1) A letter containing a detailed list of your responses to the review comments and a description of the changes you have made in the manuscript.

(2) Two versions of the manuscript: one with either highlights or tracked changes denoting where the text has been changed (uploaded as a "Revised Article with Changes Highlighted" file); the other a clean version (uploaded as the article file).

(3) If available, a striking still image (a new image if one is available or an existing one from within your manuscript). If your manuscript is accepted for publication, this image may be featured on our website. Images should ideally be high resolution, eye-catching, single panel images; where one is available, please use 'add file' at the time of resubmission and select 'striking image' as the file type. 

Please provide a short caption, including credits, uploaded as a separate "Other" file. If your image is from someone other than yourself, please ensure that the artist has read and agreed to the terms and conditions of the Creative Commons Attribution License at http://journals.plos.org/plosntds/s/content-license (NOTE: we cannot publish copyrighted images). 

(4) If applicable, we encourage you to add a list of accession numbers/ID numbers for genes and proteins mentioned in the text (these should be listed as a paragraph at the end of the manuscript). You can supply accession numbers for any database, so long as the database is publicly accessible and stable. Examples include LocusLink and SwissProt.

(5) To enhance the reproducibility of your results, we recommend that you deposit your laboratory protocols in protocols.io, where a protocol can be assigned its own identifier (DOI) such that it can be cited independently in the future. For instructions see http://journals.plos.org/plosntds/s/submission-guidelines#loc-methods

While revising your submission, please upload your figure files to the Preflight Analysis and Conversion Engine (PACE) digital diagnostic tool, https://pacev2.apexcovantage.com/ PACE helps ensure that figures meet PLOS requirements. To use PACE, you must first register as a user. Then, login and navigate to the UPLOAD tab, where you will find detailed instructions on how to use the tool. If you encounter any issues or have any questions when using PACE, please email us at figures@plos.org.

We hope to receive your revised manuscript by Feb 18 2020 11:59PM. If you anticipate any delay in its return, we ask that you let us know the expected resubmission date by replying to this email.

To submit a revision, go to https://www.editorialmanager.com/pntd/ and log in as an Author. You will see a menu item call Submission Needing Revision. You will find your submission record there. 

Sincerely,

Abdallah M. Samy, PhD

Guest Editor

Richard Phillips

Deputy Editor

Editor comments to authors: I invited and received three reviews of your manuscript. All reviews raised substantial concerns as it currently stands. I read the manuscript myself and i must say that I completely agree with the points raised by the reviewers; all points are solid and detailed. So, I would recommend major revision for your manuscript to give you an opportunity to address all the points raised by the reviewers before considering a decision on your manuscript. We cannot, of course, promise publication at that time.

Reviewer's Responses to Questions

**Key Review Criteria Required for Acceptance?**

**Methods**

-Are the objectives of the study clearly articulated with a clear testable hypothesis stated?

-Is the study design appropriate to address the stated objectives?

-Is the population clearly described and appropriate for the hypothesis being tested?

-Is the sample size sufficient to ensure adequate power to address the hypothesis being tested?

-Were correct statistical analysis used to support conclusions?

-Are there concerns about ethical or regulatory requirements being met?

Reviewer #1: These studies use a well established mouse model of M. ulcerans infection which has been a useful guide to antibiotic treatment of human infection. The methods are well applied to this study of new antibiotic combinations by authors who are familiar with the mouse model. Well designed study likely to yield clear answers. Ethical issues have been addressed.

Reviewer #2: 1) They did not show how they calculated "the M. ulcerans strain Cu001 (5.02 and 4.6 log10 in the 1st and 2nd experiment, respectively)." The right method should be calculating the CFUs from the mouse footpads on the day after infection. 

2) Line 105, "one untreated control group of 30 mice and seven treated groups" should be "eight treated groups"!

3) Line 139-141: For the untreated groups, the entire volume of the footpad suspension was plated onto 10 LJ-media with 0.2 ml each. How did the authors count the CFUs and got 5 to 7 log10CFU/footpad in some treated groups? As far as I know, it is impossible to count.

4) What is the definition of their MIC? How did the authors calculate the MICs from the CFU results? Line 160 to 162, this method to test "MIC" was wrong. 

5) “A regimen was considered to be bactericidal if its......”. This was totally wrong!

6) When the authors treat the mice in the RIF containing groups, did they separate RIF from other drugs? How long is the duration in between giving RIF and the other drug? In addition, maybe something wrong as CFUs in the RIF-STR group should be much lower than that from RIF group alone group in table 1. Can these 2 groups mixed?

7) Many (10) mice died of accident gavage and many (about 30) samples were contaminated, which indicated the perfermance/skill was not OK. Especially those in the week4 groups in table1, the samples should be diluted and so the contamination should be rare. 

8) In table 1, in the SEL group, 5 died and 1 contaminated at week4? Is it possible because of the death due to toxicity? At that time point, how about the other 10 mice for the time point week8? If they were still alive, why not kill some of them for the CFU counts at week 4 in stead of week 8？Or they died but not indicated? 

9) Why use very low dose of Q230 only at 5 mg/kg?

10) Why not test some inactive drugs in vitro first or showing the in vitro data?

Reviewer #3: See comments below

**Results**

-Does the analysis presented match the analysis plan?

-Are the results clearly and completely presented?

-Are the figures (Tables, Images) of sufficient quality for clarity?

Reviewer #1: Yes and the results, figures and tables are easily understandable.

Reviewer #2: 1)The CFU results are not so clear as some mice died or many samples were contaminated. 

2)Some results are confusing. For eg, the higher infection doses caused lower MLI. Oone possible reason is the MLI is to subjective. And CFUs in the STR-RIF group were even more than the RIF alone group. It is very abnormal. 

3)In table 2, no any mouse died in the Q203 containing groups but 3 contaminated in RIF-CLR group? For negative CFUs, all undiluted tissue should be plated, and so they can be contaminated even easier. BTW, the relapse rates were evaluated at 20 weeks after treatment stopped. The duration is a little bit shorter.

Reviewer #3: See comments below

**Conclusions**

-Are the conclusions supported by the data presented?

-Are the limitations of analysis clearly described?

-Do the authors discuss how these data can be helpful to advance our understanding of the topic under study?

-Is public health relevance addressed?

Reviewer #1: Yes to all questions.

Reviewer #2: The conclusions are OK. However, the findings that the fully intermittent oral regimens containing Q203 can cure BU in 8 weeks are not so valuable and not as good as the fully oral regimens containing Q203 which may cure BU in two weeks. And there are no anythings new except for some other drugs tested showed no activity in vivo.

Reviewer #3: See comments below

**Editorial and Data Presentation Modifications?**

Reviewer #1: A minor point is that, although the paper is easy to understand, the English is awkward in places and could do with some editing

Reviewer #2: (No Response)

Reviewer #3: See comments below

**Summary and General Comments**

Reviewer #1: These are new and important results from testing combinations of antibiotics that could well be relevant to treatment of Buruli ulcer in humans. There was a surprising improvement in time required to kill all the bacteria, particularly after Q203 containing combinations. The poor results with rifampicin plus clarithromycin (current standard of care treatment in humans) differ from previously published studies but this is discussed by the authors and their explanation is plausible. This does not detract from the value of the paper whichrepresents a valuable contribution to the literature.

Reviewer #2: The authors test several drugs including Q203 targeting QcrB, the component in the ETC. And furthermore, they found that Q203 containing fully intermittent oral regimens can cure M. ulcerans infection effectively. However, the findings are not new to the society of Buruli ulver treatment field, as cited by this manuscript, (Converse P et al, 2019 doi:10.1128/AAC.00426-19 ), it has been demonstrated that Q203 containing regimen can cure Buruli ulcer in 2 weeks. The total doses are 10 and here at least 18 doses are needed for eight weeks. So the results and conclusion in this paper is not very meaningful. In addition, not only one papers showed the cytochrome bc1-aa3 respiratory terminal oxidase inhibitors were very effective against M. ulcerans infection published in Nature Communications in 2019. 

The M. ulcerans study is very difficult and lengthy as M. ulcerans grows so slowly. So unfortunately, similar and even better results have been published when this study was going on. 

In addition, many obvious mistakes both in the methods and in experimental skills lead to the much less value of this paper.

1)It is better if the authors could provide some pictures of the swelling footpads.

2) “0.03 ml of a bacterial suspension containing around 5 log10 Colony Forming Unit (CFU) of the M. ulcerans strain Cu001 (5.02 and 4.6 log10 in the 1st and 2nd experiment, respectively.” “The mean lesion index (MLI) was 2.8 at the start of the treatment in 1 st experiment”. However, MLI was 3 at the start of the treatment in 2nd experiment. According to my experience, the higher infection dose can cause higher MLI. There should be some explanation about why a little bit lower inoculation in the 2nd animal experiment cause a little bit higher the swelling degrees of mice than that in the 1st experiment. In addition, the CFUs at the time of treatment start with higher CFUs in the 2nd experiment.

 There are some other obvious mistakes. 

3)Line 55 “.” after (WHO) should be deleted.

4)Line 76 LNZ=LZD?

5)Line 106, abbreviation should be used after it appeared in the context at the first time.

6)Line 204, P< should be italic and there should be a space in between P and <. Similar mistakes in other places.

7)Line 212, Streptomycine should be “Streptomycin”.

8) “Buruli ulcer can be successfully treated by a two-month antibiotic combination regimen administered daily” is somewhat right. However, for serious patients, the antibiotic regimens in clinical use are only as adjuvant therapies.

Reviewer #3: PNTD 19-01766

The authors first tested five drug candidates as monotherapy in the mouse footpad model of M. ulcerans infection. The most promising candidate, in terms of bactericidal activity, was Q203. This drug has a long half-life and was tested in a second experiment administered in combination with other long half-life drugs, rifapentine or bedaquiline, in an intermittent, rather than daily, dosing regimen.

The results are very interesting and potentially of great interest for the field. However, a number of major and minor concerns are noted and should be addressed. These are enumerated below.

1. How was the half-life of Q203 determined? Is there a reference for this and also for BDQ and RPT?

2. Is the MIC for PBTZ169 against M. ulcerans known? Although the sequence of DprE1 is conserved, it is not 100% identical. There could be conformational differences that would account for the significantly reduced activity. Are the MICs known for the avermectins?

3. It is an interesting question as to whether an intermittent regimen facilitates or impedes treatment completion. If dressing changes are less frequent, remembering to take the drugs twice a week could be more problematic.

4. There are a number of missing (#20-22) and also misaligned references. Please verify the connections throughout the paper. Remove editor name in PNTD refs.

5. There are disagreements between the text and the tables. For example, the text states that BDQ monotherapy was given 5/7 whereas the table states the more likely 2/7 rhythm.

6. Which MLI scale was used? Dega et al (0-4) or Lefrancois et al (0-5)?

7. It is surprising to the reviewers that the swelling values did not go below 1 in either experiment. Please use the same scale for the two figures.

8. It is suggested that for clarity the authors use color in the graphs since PNTD is an online publication.

9. Title: why “fully”? exhibit not exhibited. Thus, “Q203-containing intermittent oral regimens exhibit high sterilizing activity against Mycobacterium ulcerans in mice”.

10. Short title: remove “and” invert Q203 and intermittent. Thus, “Intermittent Q203 oral treatment against Mycobacterium ulcerans in mice”

11. Was a software program used for Mann-Whitney analysis? Please cite. Previous publications by the authors used student T-test. Why did they switch from parametric to non-parametric analysis? Adjustment for multiple comparisons is recommended. Please do chi-square or Fisher’s exact test to compare relapse proportions.

12. In Table 1, a “.” rather than a “,” should be used for the decimals. Table 2 is correct in this regard. What is the rationale for the Q203 dose selected?

13. Consistent use of abbreviations for drugs, not LNZ for LZD. The final e should not be used for selamectin, ivermectin, tedizolid, linezolid, and streptomycin in text and legends.

14. Were suspensions or supernatants of ground suspensions plated (lines 138 and 140)?

15. In the figure legends, it would be helpful to indicate whether the regimen was 5/7 or 2/7, etc.

16. There are grammar and usage errors in the English throughout. For example, when nouns are used as adjectives, the singular form of the noun must be used. Thus, mouse footpads would be correct and should be changed in many places, starting with line 38 in the abstract. Also, mouse model on line 48 and footpad lesions on line 201. Line 300 should be disappointing not deceiving (false friend in French and English). It is highly recommended that a native English speaker correct the manuscript throughout.

17. Remove “.” after “(WHO)” (line 55).

18. Tedizolid does not have higher solubility and bioavailability than linezolid. The authors may have misunderstood a statement in the reference cited that referred to improved solubility and bioavailability of the tedizolid phosphate prodrug over tedizolid (lines 75-6). 

19. “BALB/c”, not “balb/c” (line 88).

20. Where is GVKBio located (line 119)?

21. Were the lesion indexes measured weekly (line 143)?

22. “Oleic acid-Albumin-Dextrose-Catalase” (line 157).

23. Please confirm use of dimethylformamide, not dimethylsulfoxide?

24. In Table 1, there is no footnote “c” between “b” and “d”.

25. How was it determined that mice died from M. ulcerans infection (lines 220-1)?

26. In Table 2, it states that the swelling index was between 2 and 3. However, line 224 states and the figure shows 3. Please correct.

27. Table 2 is also misleading in results at 28w were not “during treatment” as suggested by the table heading.

28. In Table 2, the CFU count for Q203 2/7 at w2 has an extra “0.”

29. Line 248: “where”, not “were”

30. Please provide references for the available PK data (lines 290-291).

31. Line 298: “tedizolid” or “TDZ”, not both

32. Line 314: delete “Although”

33. Line 334: Is implantation rather than inoculum meant here? In addition, was footpad swelling also greater than usual before the onset of treatment in this experiment?

34. Line 350: probably effective rather than efficient.

35. Did the authors see any evidence of carryover of Q203 or other long half-life drugs? It may useful to readers to include a statement about the potential for drug carryover and whether it appeared to be mitigated by use of LJ.

PLOS authors have the option to publish the peer review history of their article (what does this mean?). If published, this will include your full peer review and any attached files.

Reviewer #1: No

Reviewer #2: No

Reviewer #3: No

---

## [Decision Letter · Decision Letter 1]

11 Mar 2020

Dear Mrs Chauffour,

Thank you very much for submitting your manuscript "Q203-containing intermittent oral regimens exhibit high sterilizing activity against Mycobacterium ulcerans in mice" for consideration at PLOS Neglected Tropical Diseases. As with all papers reviewed by the journal, your manuscript was reviewed by members of the editorial board and by several independent reviewers. The reviewers appreciated the attention to an important topic. Based on the reviews, we are likely to accept this manuscript for publication, providing that you modify the manuscript according to the review recommendations. 

Please prepare and submit your revised manuscript within 15 days. If you anticipate any delay, please let us know the expected resubmission date by replying to this email.  

Sincerely,

Abdallah M. Samy, PhD

Deputy Editor

Richard Phillips

Deputy Editor

Reviewer's Responses to Questions

**Key Review Criteria Required for Acceptance?**

**Methods**

-Are the objectives of the study clearly articulated with a clear testable hypothesis stated?

-Is the study design appropriate to address the stated objectives?

-Is the population clearly described and appropriate for the hypothesis being tested?

-Is the sample size sufficient to ensure adequate power to address the hypothesis being tested?

-Were correct statistical analysis used to support conclusions?

-Are there concerns about ethical or regulatory requirements being met?

Reviewer #1: Please see previous report

Reviewer #3: The methods are now acceptable for this reviewer.

Reviewer #4: This is a well-designed, carefully performed animal experiment allowing for straightforward conclusions to be drawn; the number of experimental animals have been accounted for, and is likely to be justified to be able to reach at the conclusions; the paper adds to what we know about Q203 (why not use the new name telacebec?) and many questions regarding the potential benefit of intermittent therapy have been addressed, carrying out many control experiments in the same model. All failed experiments have been accounted for, making the paper very convincingly robust.

Reviewer #5: The objectives of the study are clearly articulated with a clear testable hypothesis stated. Also, the study design was appropriate to address the stated objectives. 

The population was clearly described and appropriate for the hypothesis being tested, and the sample size was sufficient to ensure adequate power to address the hypothesis being tested. Correct statistical analysis were used to support conclusions but specific p values were not presented. Concerns about ethical or regulatory requirements were met. Below are some specific statements. 

1. Rewrite lines 103 – 104 to read: The mice were randomly allocated into eight groups (1st experiment) and ten groups (2nd experiment) using a randomization table generated by the web site Randomization.com (http://www.randomization.com). 

2. Lines 106 and 110. Write “1st experiment” as “First experiment” and 2nd Experiment as “Second experiment” since they are beginning the sentence. 

3. Line 105: Please what do you mean by: The groups were as follows (drug, dosage, number of doses/week)???

4. Lines 178 – 179: Please provide a reference for your MIC definition. 

5. Line 197: Heading should be “First experiment” instead of 1st experiment

6. Why are you not providing the specific p values? You mostly quoted P values as (p <0.05) line 217 and 271, (p < 0.001) line 225, (p ≤0.01), line 267 and (p <0.002) line 282. Please try and provide the specific P values, E.g (p = 0.012).

**Results**

-Does the analysis presented match the analysis plan?

-Are the results clearly and completely presented?

-Are the figures (Tables, Images) of sufficient quality for clarity?

Reviewer #1: Yes

Reviewer #3: Fig. 1: The reviewer finds (A) lesion index 1 not typical of grade 1swelling; more like, 0-1, improving after treatment. Other images for grades 2, 3, and 4 are fine.

Line 139: why not 0-1?

Table 1: Please indicate 2.8± SD at start of treatment

Line 245: 3±SD

Line 265: the CFUs were not unchanged but you could say stable

Reviewer #4: results are well presented, and graphics and tables are clear.

Reviewer #5: 1. The analysis presented matched the analysis plan. The results were clearly and completely presented but the figures (Images) are not of sufficient quality for clarity. Quality of the figures need to be improved. 

2. Figure 1: Authors need to calculate and show the image sizes to help with comparison of the Lesion index grades

3. Figures 2 and 3: On the Y-axis, authors have use comma (,) instead of point (.). Please correct throughout. E.g. 1,0, 1,5, and 2,0, should be 1.0, 1.5 and 2.0.

**Conclusions**

-Are the conclusions supported by the data presented?

-Are the limitations of analysis clearly described?

-Do the authors discuss how these data can be helpful to advance our understanding of the topic under study?

-Is public health relevance addressed?

Reviewer #1: Yes

Reviewer #3: Line 325: Are mice good animal models for avermectins? These drugs are very potent at low doses in humans for parasites and scabies mites but much higher doses are needed in mice and may be toxic.

Line 358: The authors might note that relapse occurred at similar proportions in other mouse studies:

Almeida et al. after 4 weeks treatment in a kinetic rather than curative model, PMID: 21245920

Converse et al. after 8 weeks treatment in a curative model, Figs 3& 4, PMID: 26042792

Converse et al. after 6 weeks treatment in a curative model, Fig. 2, PMID: 30102705

Reviewer #4: conclusions are all supported by the data presented; limitations (especially, using high inoculums resulting in persisting live bacteria after 8 wk treatment) have been addressed; the fact that only one wild-type M ulcerans isolate was used was not mentioned as a potential weakness, but this does not seem a very important limitation anyway; the authors have a clear mindset to help advance treatment of Buruli ulcer in rural Africa; and they help the reader how to understand some of the scientific questions regarding drug treatment of M ulcerans infection.

Reviewer #5: 1. Line 336: The authors indicated that RIF and RIF-STR were highly bactericidal as in all their preceding works [20]. Meanwhile they cited just one publication (Ref 20).

2. The authors tested several drugs including Q203 (now named telacebec) and thus observed that Q203-containing intermittent oral regimens can cure M. ulcerans infection effectively. However, the findings are not novel with regards to Buruli ulcer treatment. A publication by Converse et al, 2019 (doi:10.1128/AAC.00426-19) observed that footpad swelling decreased more rapidly in mice treated with Q203-containing regimens than in mice treated with RIF and STR (RIF+STR) and RPT and CFZ (RPT+CFZ). In that study nearly all footpads were culture negative after only 2 weeks of treatment with regimens containing RPT, CFZ, and Q203. No relapse was detected after only 2 weeks of treatment in mice treated with any of the Q203-containing regimens. In contrast, 15% of mice receiving RIF+STR for 4 weeks relapsed. Converse et al (2019) concluded that it may be possible to cure patients with Buruli ulcer in 14 days or less using Q203-containing regimens rather than currently recommended 56-day regimens.

In the current study under review, as mentioned earlier, the authors evaluated the bactericidal activity of several new antimicrobials drugs in a mouse model of BU and found that the Q203 exhibited the highest bactericidal effect. They also subsequently identified new antibiotic combinations containing Q203 with high sterilizing activity when administrated twice a week for 8 weeks, i.e. for a total of only 16 doses. The authors need to highlight the latter more and indicate the new combinations to bring out the novelty of this study rather than portraying the already established idea that Q203-containing intermittent oral regimens exhibit high sterilizing activity against Mycobacterium ulcerans in mice. Thus, I recommend a modification to the title and key findings should also be based more on the new Q203-containing combinations identified. 

REF: 

Converse, P. J., Almeida, D. V., Tyagi, S., Xu, J., & Nuermberger, E. L. (2019). Shortening Buruli Ulcer Treatment with Combination Therapy Targeting the Respiratory Chain and Exploiting Mycobacterium ulcerans Gene Decay. Antimicrobial agents and chemotherapy, 63(7), e00426-19. https://doi.org/10.1128/AAC.00426-19

3. The study by Converse et al, 2019 is very instrumental with regards to the current study under review; meanwhile, the authors cited that study just once (at the last few sentences of the discussion). There are a number of areas in the earlier sections that this reference needed to be mentioned. 

4. Lines 371 -381: Authors need to give a more detailed contrast of the current study and that of Converse et al, 2019. What has been indicated there is not enough. 

5. Lines 379 -380: “thanks to the long half-life and good bio availabilities of these drugs” needs to be re-written as it is not a proper way of scientific writing. 

6. Generally, the conclusions based on the major findings are OK, however, I am not so sure of the novelty of major findings highlighted by the authors.

**Editorial and Data Presentation Modifications?**

Reviewer #1: Minor revision of the use of English

Reviewer #3: Abstract

Line 31: delete “the”

Line 34: (an oxazolidinone …) … (avermectin compounds)

Author summary

Line 51: the verb form should be administered. Please change throughout manuscript. Or, administering, as appropriate.

Introduction

Line 61: patient adherence 

Lines 82 and 85: Please cite Pethe et al. then Rosenthal et al and Rouan et al here, too.

Methods

Line 88: 4 week-old 

Line 119: Is this the correct affiliation for Dr. Cole?

Line 129: in the same volume

Line 142: ground in …… in a final

Line 160: 8 week

Line 162 and 163: for instead of during

Line 171: McFarland

Line 187 and 188: Ethics/ethics. The experiment project

Results

Line 208: Line 258: Fig 2. First experiment: Change in mean lesion index of M. ulcerans-infected mouse footpads during treatment for 8 weeks

Line 217: mouse group

Line 222: lesions

Table 1

b,e,i: advanced lesions

 c and d: due a gavage accident

line 246: swelled

Fig 3

Line 258: Fig 3. Second experiment: Change in mean lesion index of M. ulcerans-infected mouse footpads during and after treatment for 8 weeks

Line 260: Dosages were as follows:

Line 267: the untreated group.

Table 2

a: treatment began

c,e: advanced lesions

e,f: due a gavage accident

Discussion:

Line 322: The same

Line 326: humans

Line 349: administering twice weekly for 8 weeks (similar correction line 352-353)

Line 356: with low CFU counts

Line 368: The reason for this

Line 371: italicize bc and aa

Line 380: bioavailability

I suggest that the first line of the Discussion section should be changed to reflect the preponderance of current expert opinion on Buruli ulcer treatment. Therefore, it should be changed accordingly:

“Although nearly all Buruli ulcer lesions can be successfully treated by a two-month antibiotic combination regimen administered daily, …”

The authors may wish to cite Wadagni et al. PMID: 31658295 and/or PMID: 29605498 as well as Clinicaltrials.gov: NCT01659437. I believe the latter study may (soon) be in press in the Lancet.

Reviewer #4: This is good science, presented in a clear fashion; and the only thing I see as something that might be improved is the use of prepositions and the English language article 'the' but it does not interfere with the clarity of the text.

Reviewer #5: 1. The authors indicated that “Buruli ulcer (BU), caused by Mycobacterium ulcerans, was only treated by surgery until 2004”. Is this on a worldwide scale? If Yes, please indicate. Also, this statement sounds too emphatic and thus would require reference(s) to support, as done for the statement which showed the first medical treatment of Buruli ulcer recommended by the World Health Organization.

2. Lines 64 – 67: Please provide references for the stamen below:

For instance, many Buruli ulcer patients with small-to-moderate size wounds are on

ambulatory care [REF], and visit healthcare centres twice or three times per week for dressing changes [REF], a rhythm that could allow receiving supervised intermittent antibiotic administration.

3. The entire manuscript needs English Grammar editing and proofreading. E.g. Line 88. There should be a comma (,) after the word “Respectively”. Also, line 88 -89, since 1st and 2nd were appearing for the first time they need to be written in full as First (1st ) and second (2nd) experiments.

**Summary and General Comments**

Reviewer #1: See below

Reviewer #3: The manuscript is much improved but there are editorial issues remaining that should have been addressed in the first revision.

Line 306: I am not sure that successful treatment with antibiotics is restricted to small-to-moderate lesions. Even edematous lesions have been reported to be cured in the various clinical trials.

Reviewer #4: I personally like a Discussion section starting with the most striking finding from the work presented ('this study provides evidence of excellent healing of M ulcerans infection without relapse in the mouse model, using the novel agent telacebec in a twice-weekly regimen for only 8 weeks'. But, as this is the report on an animal experiment, perhaps the current Discussion is good enough for readers that are used to read papers on experimental research. PLoS NTD has however also a clinical readership; no doubt, clinicians pick up the important message from the title as well.

Reviewer #5: The current study used a well-established mouse model of M. ulcerans infection which can be considered a useful guide to treatment of human infection. Ethical issues have been adequately 

addressed. However, there are several major issues which need to be addressed before this manuscript can be considered for publication. Also, I am not so sure of the novelty of major findings highlighted by the authors and significance of the study was not highlighted very much in the writing.

PLOS authors have the option to publish the peer review history of their article (what does this mean?). If published, this will include your full peer review and any attached files.

Reviewer #1: No

Reviewer #3: No

Reviewer #4: No

Reviewer #5: No
---

## [Editor Report · Decision Letter 2]

15 Jun 2020

Dear Mrs Chauffour,

We are pleased to inform you that your manuscript 'Telacebec (Q203)-containing intermittent oral regimens sterilized mice infected with Mycobacterium ulcerans after only 16 doses' has been provisionally accepted for publication in PLOS Neglected Tropical Diseases.

Best regards,

Abdallah M. Samy, PhD

Deputy Editor

Richard Phillips

Deputy Editor

---

## [Editor Report · Acceptance letter]

28 Jul 2020

Dear Mrs Chauffour,

We are delighted to inform you that your manuscript, "Telacebec (Q203)-containing intermittent oral regimens sterilized mice infected with Mycobacterium ulcerans after only 16 doses," has been formally accepted for publication in PLOS Neglected Tropical Diseases.

Best regards,

Shaden Kamhawi

co-Editor-in-Chief

Paul Brindley

co-Editor-in-Chief
